



**Last ice sheet recession and landscape emergence above sea level in east-central Sweden, evaluated**
**using *in situ cosmogenic* [14]C from quartz**
Bradley W. Goodfellow[1*]
Arjen P. Stroeven[2,3]
Nathaniel A. Lifton[4,5]
Jakob Heyman[6]
Alexander Lewerentz[1]
Kristina Hippe[7]
Jens-Ove Näslund[8]
Marc W. Caffee[4,5]
[1]Geological Survey of Sweden
[2]Department of Physical Geography, Stockholm University
[3]Bolin Centre for Climate Research, Stockholm University
[4]Department of Earth, Atmospheric, and Planetary Sciences, Purdue University
[5]Department of Physics and Astronomy, Purdue University
[6]Department of Earth Sciences, University of Gothenburg
[7]Umweltplanung Dr. Klimsa
[8]Swedish Nuclear Fuel and Waste Management Company (SKB)
*Corresponding author: bradley.goodfellow@sgu.se

**Abstract**
*In situ* [14]C in quartz provides a recently developed tool to date exposure of bedrock surfaces up to
~25 000 years. From outcrops located in east-central Sweden, we test the accuracy of *in situ* [14]C dating
against (i) a relative sea level (RSL) curve constructed from radiocarbon dating of organic material in
isolation basins, and (ii) the timing of local deglaciation constructed from a clay varve chronology
complemented with radiocarbon dating. Five samples of granitoid bedrock were taken along an
elevation transect extending southwestwards from the Baltic Sea coast near Forsmark. Because these
samples derive from bedrock outcrops positioned below the highest postglacial shoreline, they target
the timing of progressive landscape emergence above sea level. In contrast, *in situ* [14]C concentrations
in an additional five samples taken from granitoid outcrops above the highest postglacial shoreline,
located 100 km west of Forsmark, should reflect local deglaciation ages. The ten *in situ* [14]C



measurements provide robust age constraints that, within uncertainties, compare favorably with the
RSL curve and with the local deglaciation chronology. These data demonstrate the utility of *in situ* [14]C
to accurately date ice sheet deglaciation, and durations of postglacial exposure, in regions where
cosmogenic [10]Be and [26]Al routinely return complex exposure results.
**1. Introduction**
The pacing of retreat of ice sheets in North America and Eurasia since their maximum expansion
during the last glaciation remains an active research field (e.g., Hughes et al., 2016; Stroeven et al.,
2016; Patton et al., 2017; Dalton et al., 2020, 2023). Understanding the triggers and processes causing
the demise of these ephemeral ice sheets yields the best blueprint for understanding the future
behavior of the Greenland and Antarctic ice sheets in a warming climate. Coupling the behavior of
deglaciating ice sheets over the course of the Late Glacial and early Holocene to increasingly precise
climate reconstructions and climatic events, requires increased precision in ice sheet reconstructions
(e.g., Bradwell et al., 2021). Increased precision can be achieved through a coupling of
geomorphological mapping of ice sheet margins (such as moraines, grounding zone wedges, lateral
meltwater channels, and ice-dammed lake shorelines and spillways) with numerical field constraints
from a diverse array of dating techniques (e.g., Stroeven et al., 2016; Bradwell et al., 2021; Regnéll et
al., 2023).
Ice sheet reconstructions, especially in North America, have attained a high level of detail through
radiocarbon dating (Dyke et al., 2002; Dalton et al., 2020). With the advance of offshore imaging of
glacial geomorphology (Greenwood et al., 2017, 2021; Bradwell et al., 2021), radiocarbon dating has
received a renewed upswing in recent years (e.g., Dalton et al., 2020; Bradwell et al., 2021). However,
large tracts of landscape lack radiocarbon age constraints on ice sheet retreat simply due to a lack of
datable organic material. Fortunately, optically-stimulated luminescence ages on buried sand layers
(e.g., Alexanderson et al., 2022) and cosmogenic nuclide apparent exposure ages on exposed bedrock
and erratics have narrowed some of the gaps (e.g., Hughes et al., 2016; Stroeven et al., 2016; Dalton et
al., 2023). In studies using cosmogenic nuclides, an 'apparent' exposure age is derived from a simple
calculation from the nuclide concentration under consideration (Lal, 1991; Gosse and Phillips, 2001).
However, correctly interpreting the exposure age relies on modelling that considers geological factors
that can reduce the nuclide concentration relative to the time since initial subaerial exposure (such as
erosion and burial by glacial ice, water, snow, and/or soil; Gosse and Phillips, 2001; Schildgen et al.,
2005; Ivy-Ochs and Kober, 2008). Exposure dating is the only technique available in regions where ice
sheet erosion has left the surface bare or covered by a thin drape of till. Kleman et al. (2008) show that
for Fennoscandia, these conditions are widespread in coastal regions where ice accelerated towards its



streaming sectors and where wave wash during glacial rebound further thinned or removed pre-
existing sediment covers.
Coastal sectors in formerly glaciated regions provide sites important to the study of paleoglaciology.
They offer an abundance of bedrock exposures from which patterns and processes of subglacial
erosion can be studied through cosmogenic nuclide exposure dating (e.g., Hall et al., 2020). Also,
because of the interplay with postglacial sea level, coastal areas yield data on glacioisostatic rebound
that are critical to geodynamic modelling of Earth rheology and thicknesses of former ice sheets (e.g.,
Lambeck et al. (1998, 2010) and Patton et al. (2017), for Fennoscandian examples). Geodynamic
models require validation against measurements of vertical crustal motion (Steffen and Wu, 2011),
such as those provided by recent global positioning system (GPS) measurements (e.g., Lidberg et al.,
2010) and postglacial records of crustal rebound afforded by relative sea level (RSL) curves (e.g., Påsse
and Andersson, 2005). The construction of RSL curves, detailing the history of land surface emergence
from sea level, is traditionally done using either sediments accumulated in isolation basins at different
elevations above sea level or by dating uplifted gravel beach ridges. Typically, isolation basins, and their
sediments, show a progression from marine, to brackish, and finally to freshwater environments as
their bedrock sills are uplifted through tidal levels (Long et al., 2011). Histories of land uplift above sea
level are documented using micro- and macrofossil analyses of isolation basin sediments and
radiocarbon dating on macrofossils (Romundset et al., 2011). Uplifted beach ridges can be radiocarbon
dated from a variety of materials (Blake, 1993) but most confidently from driftwood, whalebone, and
shells (e.g., Dyke et al., 1992). Gravel beach ridges have also been investigated using OSL and $^{10}$Be
exposure dating even though, other than the highest beach ridge, they may be prone to clast
reworking (Briner et al., 2006; Simkins et al., 2013; Bierman et al., 2018). A distinct advantage of
constructing RSL curves using cosmogenic nuclides is that land surface emergence above sea level may
be additionally dated from boulders (Briner et al., 2006) or bedrock (Bierman et al., 2018).
The potential for cosmogenic surface exposure dating of last ice sheet retreat in recently glaciated low-
relief cratonic landscapes would seemingly be high because of the frequent outcropping of glacially
sculptured quartz-bearing crystalline bedrock. However, the ice sheet may have been either non-
erosive or erosion was insufficiently deep to remove all the cosmogenic nuclide inventory from
previous exposure periods. Apparent ages are therefore often older than indicated by radiocarbon
dating (Heyman et al., 2011; Stroeven et al., 2016) because they include a component of nuclide
inheritance. Apparent ages younger than indicated by radiocarbon dating can also occur if sampled
rock surfaces have been shielded, for example by sediments, following deglaciation. Concentrations of
$^{10}$Be and $^{26}$Al, in either bedrock or erratic boulders, therefore often reflect complex exposure histories
rather than simple deglacial exposure durations (Heyman et al., 2011; Stroeven et al., 2016).



In this study we use $^{14}$C produced *in situ* in quartz-bearing bedrock (*in situ* $^{14}$C) because it potentially
circumvents an overt reliance on the need for deep erosion (> 3 m) to remove the inherited signal from
previous exposure periods (Gosse and Phillips, 2001). The reason for this is that, because of its short
half-life of 5700 ± 30 years, nuclide inheritance will have largely decayed away if ice sheet burial at
investigated sites during the last glacial phase (marine isotope stage 2; MIS2) exceeded 25-30 ka, that
is, ca. 5 half-lives (Briner et al., 2014).
Some studies assessing changes in glacier and ice sheet extents over Late Glacial to Holocene
timescales have used *in situ* $^{14}$C (Miller et al., 2006; Fogwill et al., 2014; Hippe et al., 2014;
Schweinsberg et al., 2018; Pendleton et al., 2019; Young et al., 2021; Schimmelpfennig et al., 2022). In
such studies, *in situ* $^{14}$C has been applied with other nuclides with longer half-lives, in particular $^{10}$Be,
to unravel complex histories of glacier advance and retreat (e.g., Goehring et al., 2011) and spatial
patterns in glacial erosion in mountainous terrain (e.g., Steinemann et al., 2021). However, extensive
regions formerly covered by ice sheets are characterized by low relief, low elevation terrain, and the
effectiveness of *in situ* $^{14}$C in dating ice sheet retreat in these non-alpine settings and in quantifying
shoreline displacement from bedrock samples has not been previously assessed. The aim of this study
is therefore to validate the use of $^{14}$C formed *in situ* in bedrock as a reliable chronometer by evaluating
its performance in duplicating (i) a previously-established Holocene RSL curve based on radiocarbon
dating (Hedenström and Risberg, 2003; SKB, 2020) and (ii) the timing of deglaciation above the highest
(post-glacial) shoreline in nearby east-central Sweden according to reconstructions of deglaciation of
the last ice sheet (Hughes et al., 2016; Stroeven et al., 2016).

**2. Study Area**
Our study is focused on a region that includes low elevation, low relief, Forsmark-Uppland and
adjoining higher elevation and relief Dalarna-Gävleborg in east-central Sweden (Fig. 1). This region was
selected because Forsmark is the location of a planned geological repository for spent nuclear fuel
(e.g., SKB 2022) and therefore also has abundant geologic data relevant to our study. This includes in-
depth analyses of bedrock and environmental properties, including influences of glacial and postglacial
processes (e.g., Lönnqvist and Hökmark, 2013; Hall et al., 2019; Moon et al., 2020; SKB, 2020).
From spatio-temporal ice sheet reconstructions by Kleman et al. (2008), the study area was glaciated
16-20 times for a total duration of c. 330 ky over the past 1 Ma. The last deglaciation of the study area
is well-constrained by two recent reconstructions that differ in their approach (Hughes et al., 2016;
Stroeven et al., 2016). The Hughes et al. (2016) reconstruction is explicitly based on chronological
constraints, but the Stroeven et al. (2016) reconstruction combines geomorphological constraints for
ice sheet margin outlines with chronological constraints. Whereas Hughes et al. (2016) reconstruct ice



sheet retreat every 1 ka, and for every ice margin plot its position as "most credible", "min", and
"max", Stroeven et al. (2016) present ice margin positions for every 100 years inside the Younger Dryas
standstill position (Stroeven et al., 2015). These marginal positions are temporally and spatially defined
by the "Swedish Time Scale" clay varve record along the Swedish east coast (De Geer, 1935, 1940;
Strömberg, 1989, 1994; Brunnberg, 1995; Wohlfarth et al., 1995). From Stroeven et al. (2016), the last
deglaciation of the study area occurred 10.8 ± 0.3 ka BP, which overlaps the timing of deglaciation of
the study area from Hughes et al. (2016), within uncertainty (Fig. 1). The highest postglacial shoreline
in east-central Sweden is located at a present elevation of ~200 m a.s.l. in Dalarna-Gävleborg, ~100 km
west of Forsmark (SGU, 2015). The exposure duration of bedrock above the highest postglacial
shoreline therefore represents the time since local deglaciation. Hence, *in situ* $^{14}$C ages from bedrock
above the highest postglacial shoreline should conform to the reconstructed deglaciation age of 10.8 ±
0.3 ka from Stroeven et al. (2016).
Below the highest postglacial shoreline, in the Forsmark-Uppland region, the last deglaciation
occurred in a marine environment and the landscape has progressively emerged above sea level
through postglacial isostatic uplift. A RSL curve constructed from radiocarbon dating of basal organic
sediments trapped in isolation basins along elevation transects describes the progressive emergence
of the Forsmark-Uppland landscape above sea level (Robertsson and Persson, 1989; Risberg, 1999;
Bergström, 2001; Hedenström and Risberg, 2003; Berglund, 2005; SKB, 2020). Ages calculated from *in*
*situ* $^{14}$C from bedrock outcrops along an elevation transect would then mirror the Forsmark RSL curve
for their corresponding elevations (but be slightly older because of nuclide production through
shallow water before emergence).
A potential complication to the accurate exposure age dating of bedrock surfaces using *in situ* $^{14}$C in
east-central Sweden is that the most recent period of ice sheet burial may not have been sufficiently
long to decay the *in situ* $^{14}$C inventory inherited from preceding exposure. Here, the extent of the
Fennoscandian Ice Sheet during interstadial MIS3 and the timing of ice advance across the Forsmark
region during late MIS3 are crucially important. Kleman et al. (2020) have identified ice-free conditions
around Idre (330 km NW, up-ice, of our study area; Fig. 1) between 55 ka and 35 ka, which implies
inundation of our study area by ice after 35 ka. Combined with a well-constrained final deglaciation
age of 10.8±0.3 ka (Stroeven et al. 2016), it appears that our study area has most recently (during
MIS2) been inundated by glacial ice for at most 24 ka. This inference is in line with results from ice
sheet modelling indicating a 22 kyr duration of ice-cover at Forsmark during MIS2 (SKB, 2020).
Consequently, it is possible that *in situ* $^{14}$C concentrations may reflect subaerial exposure of bedrock in
our study area during MIS3 in addition to Holocene exposure, resulting in an offset towards older ages
relative to the RSL curve for Forsmark (Hedenström and Risberg, 2003; SKB, 2020) and the deglaciation
chronologies of Hughes et al. (2016) and Stroeven et al. (2016).




## 3. Methods

### 3.1. Sampling of bedrock outcrops for *in situ* [14]C measurement

We used the following sampling strategy to evaluate the accuracy of bedrock exposure ages derived
from *in situ* [14]C against the Forsmark RSL curve and the deglaciation of the last ice sheet in east-central
Sweden. A rigorous scheme was applied to ensure that we avoided sampling quartz altered through
hydrothermal processes that is likely to occur in major pegmatite intrusions, outcrops located in major
deformation zones, and outcrop-scale veins, fractures, and adjacent rock volumes. Consequently,
sampling was done on outcrops of metagranitoid from the early-Svecokarelian GDG-GSDG suite that
dominates the Bergslagen lithotectonic unit (Stephens and Jansson, 2020). A petrological examination
using transmitted light polarization microscopy was applied to thin sections to ascertain that the quartz
was unlikely to contain multi-fluid phase, vapour phase, or solid-phase inclusions. All samples were
collected using an angle grinder, which permits sampling of hard crystalline bedrock isolated from
outcrop edges, fractures, and quartz veins, and consistently limits sample thicknesses to 3 cm.
We collected a total of ten samples for *in situ* [14]C analyses. Five of these were collected along a SW-NE
transect near Forsmark (Fig. 1b). These outcrops were chosen because they span an elevation gradient
of 9.4–56.0 m a.s.l. and exposure ages derived from *in situ* [14]C can therefore be evaluated against the
Forsmark RSL curve. We collected a further five samples from locations above the highest shoreline (Fig.
1a) to determine the age of local deglaciation for comparison with published deglaciation chronologies
(Hughes et al., 2016; Stroeven et al., 2016). Sample locations were logged on a 2 m-resolution LiDAR
digital elevation model (DEM) displayed in ArcGIS 10 on a tablet computer. A GPS add-in tool in ArcGIS
10 was used to record positional data, within a horizontal precision of 2 m. The elevation of each sample
location was extracted from the DEM and has a precision of tens of centimetres. The influence of these
minor positional uncertainties on our [14]C calculations is trivial and none of the sample sites is influenced
by topographic shielding that could reduce the accumulation of [14]C in bedrock.
Each sampled bedrock outcrop formed a local topographic high, which minimizes the risk of burial by
soil and snow (Supplement 1). Moss mats were present on all sampled outcrops. Although we avoided
sampling bedrock that was moss-covered, we cannot be certain that moss mats did not formerly cover
the sample sites. Given a compressed thickness of 0.5 cm and an estimated density of 0.7 g/cm$^3$, this
may have contributed to a shielding of the sampled rock surfaces of 0.35 g/cm$^2$, which is negligible and
is therefore excluded from our age inferences.



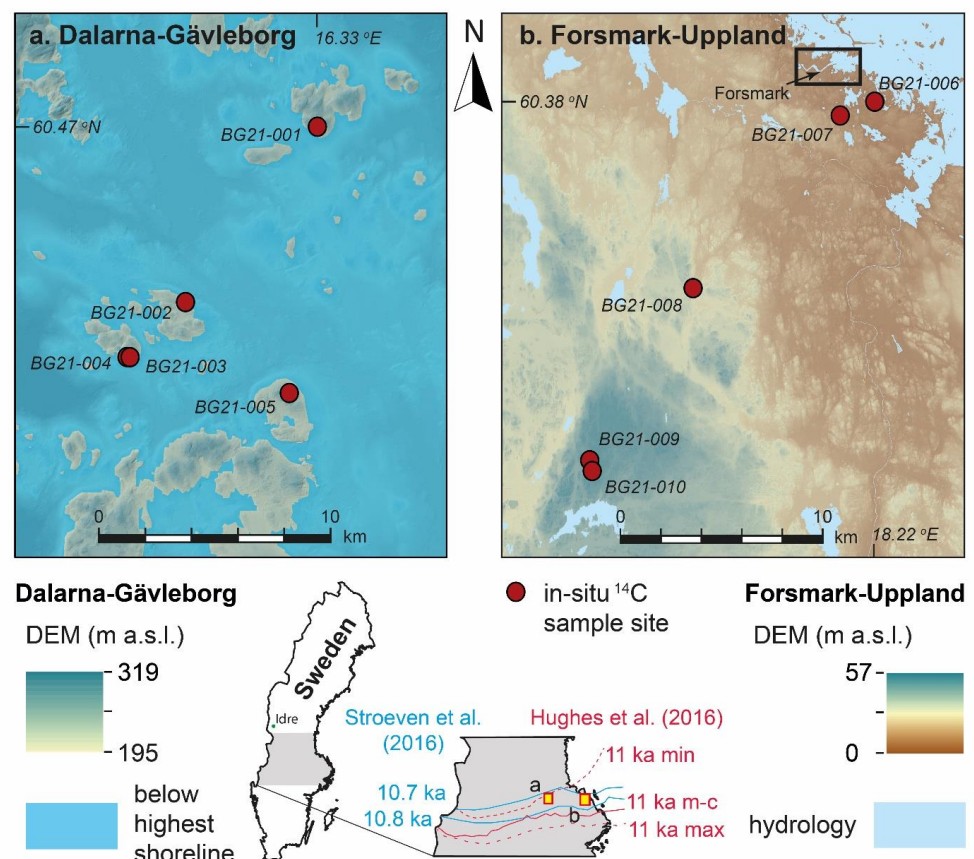

**Figure 1**. Sample locations for *in situ* [14]C dating in **(a)** Dalarna-Gävleborg and **(b)** Forsmark-Uppland. The five Dalarna-Gävleborg sample sites are located above the highest postglacial shoreline (shown), whereas the five sample sites from Forsmark-Uppland are located below the highest shoreline (not shown because the entire area was submerged). See inset maps for locations of panels a and b and for the 10.7 ka BP and 10.8 ka BP retreat isochrones (blue) from Stroeven et al. (2016) and 11 ka BP (most-credible, minimum, and maximum) retreat isochrones (red) from Hughes et al. (2016). The rectangle in panel b approximately indicates the site selected for the planned geological repository for spent nuclear fuel at Forsmark. DEM with 2 m resolution, from LiDAR data, Lantmäteriet.

### 3.2. Laboratory preparation for accelerator mass spectrometry (AMS)

Samples were physically and chemically processed at the Purdue Rare Isotope Measurement Laboratory (PRIME Lab) at Purdue University, U.S.A. Concentrations of *in situ* [14]C were determined from purified quartz separates through automated procedures (Lifton et al., 2023). Approximately 5 g of quartz from each sample was added to a degassed $LiBO_2$ flux in a re-usable 90% Pt/10% Rh sample boat and heated



to 500 °C for one hour in ca. 6.7 kPa of Research Purity $O_2$ to remove atmospheric contaminants, which
were discarded. The sample was then heated to 1100 °C for three hours to dissolve the quartz and
release the *in situ* [14]C, again in an atmosphere of ca. 6.7 kPa of Research Purity $O_2$ to oxidize any evolved
carbon species to $CO_2$. The $CO_2$ from the 1100 °C step was then purified, measured quantitatively, and
converted to graphite for [14]C AMS measurement at PRIME Lab (Lifton et al., 2023). To test for data
reproducibility, sample BG21-002 was randomly selected to undergo laboratory preparation and AMS a
second time. Measured concentrations of *in situ* [14]C are calculated from the measured isotope ratios via
AMS following Hippe and Lifton (2014).

### 3.3. Exposure age calculations

The expage calculator version 202312 (http://expage.github.io/calculator) is used to calculate apparent
exposure ages. It is based on the original CRONUS calculator v. 2 (Balco et al., 2008), the LSDn production
rate scaling (Lifton et al., 2014), and the CRONUScalc calculator (Marrero et al., 2016), using the
geomagnetic framework of Lifton (2016) with the SHA.DIF.14k model for the last 14 kyr. Exposure ages
are calculated using resulting time-varying [14]C production rates accounting for decay and interpolated
to match the measured [14]C concentration. The production rate from muons is calibrated against the
Leymon High core [14]C data of Lupker et al. (2015) and the production rate from spallation is calibrated
against updated global [14]C production rate calibration data (Schimmelpfennig et al., 2012; Young et al.,
2014; Lifton et al., 2015; Borchers et al., 2016; Phillips et al., 2016; Koester and Lifton, 2023). This
calibration is done iteratively for spallation and muons to reach convergence, using the expage
production rate calibration methods (Fig. 2).
Exposure age calculations along the Forsmark-Uppland transect account for [14]C production during
emergence through shallow water. However, burial of sampled surfaces by snow is excluded from the
age calculations for all sample sites because we neither know how snow burial depths and durations
vary between sites nor vary through time. The effect of snow burial would be to slightly decrease
cosmogenic nuclide production in the underlying rock surface (Schildgen et al., 2005) and we have
minimized this effect through our sampling strategy.



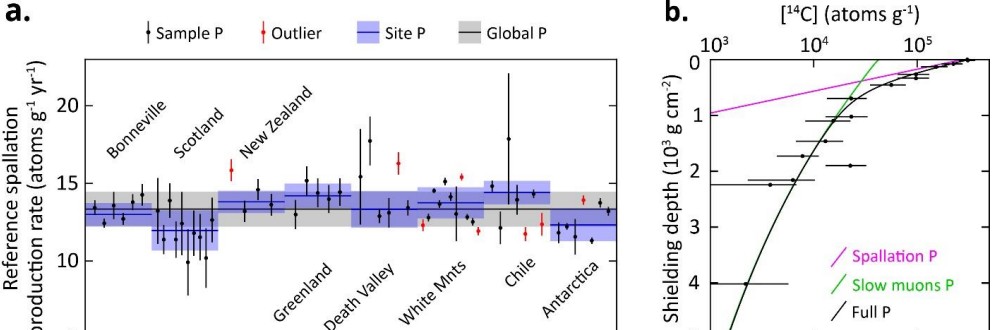

**Figure 2**. Production rate calibration of $^{14}$C in quartz. **(a)** Reference spallation $^{14}$C production rate
calibration based on data from Schimmelpfennig et al. (2012), Young et al. (2014), Lifton et al. (2015),
Borchers et al. (2016), and Phillips et al. (2016), corrected per Hippe and Lifton (2014) and compiled in
Koester and Lifton (2023). An uncertainty-weighted production rate is calculated for each of the eight
sites. Outliers, which are not included in the uncertainty-weighted production rates, are determined
based on the requirement that there should be at least three samples yielding a reduced chi-square
statistic ($X_R^2$) with a p-value of at least 0.05 for the assumption that the individual production rates from
a site are derived from one normal distribution. For $X_R^2$, but not the uncertainty-weighting, we use the
largest of the sample-specific production rate uncertainty based on the $^{14}$C concentration uncertainties
and 5% of the sample production rate. This procedure does not punish samples with low measurement
uncertainties, which otherwise risk exclusion as outliers. We adopt a global reference spallation $^{14}$C
production rate of 13.35 ± 1.13 atoms g$^{-1}$ yr$^{-1}$, calculated as the arithmetic mean of the eight site
production rates with the uncertainty being based on an uncertainty-weighted deviation of all included
single sample production rates, excluding outliers. **(b)** Calibration of $^{14}$C production rate from muons
based on the data of Lupker et al. (2015). The calibration is based on the method used in the CRONUScalc
calculator (Marrero et al., 2016; Phillips et al., 2016). The figure shows the best fit $^{14}$C concentration
profiles produced from spallation, slow muons, and full production. The best fit yields near zero
production from fast muons (cf. Lupker et al., 2015). The production rate calibration has been carried
out using the expage-202306 calculator in an iterative way to make the global reference spallation $^{14}$C
production rate converge with the production rate from muons.

**4. Results**
Inferred ages for the five *in situ* $^{14}$C samples from the Forsmark-Uppland transect (i.e., below the highest
postglacial shoreline) are shown relative to the Holocene RSL curve for Forsmark and the expected *in*
*situ* $^{14}$C exposure age curve considering subaqueous cosmogenic nuclide production (Figure 3; Tables 1
and 2). Exposure age uncertainties are large with internal uncertainties (measurement uncertainties;



Balco et al., 2008) of 5-9% and external uncertainties of 12-20% (also including production rate
uncertainties, which are high relative to [10]Be (Borchers et al., 2016; Phillips et al., 2016). Apparent
exposure ages increase consistently with elevation and match expected ages within uncertainty. The two
highest samples have near-identical apparent exposure ages and elevations. However, these samples
provide independent ages because they are horizontally separated by 624 m (Figure 1b). There is good
agreement between ages inferred from these *in situ* [14]C data and the RSL curve constructed from organic
radiocarbon dating of isolation events (Hedenström and Risberg, 2003; SKB, 2020).

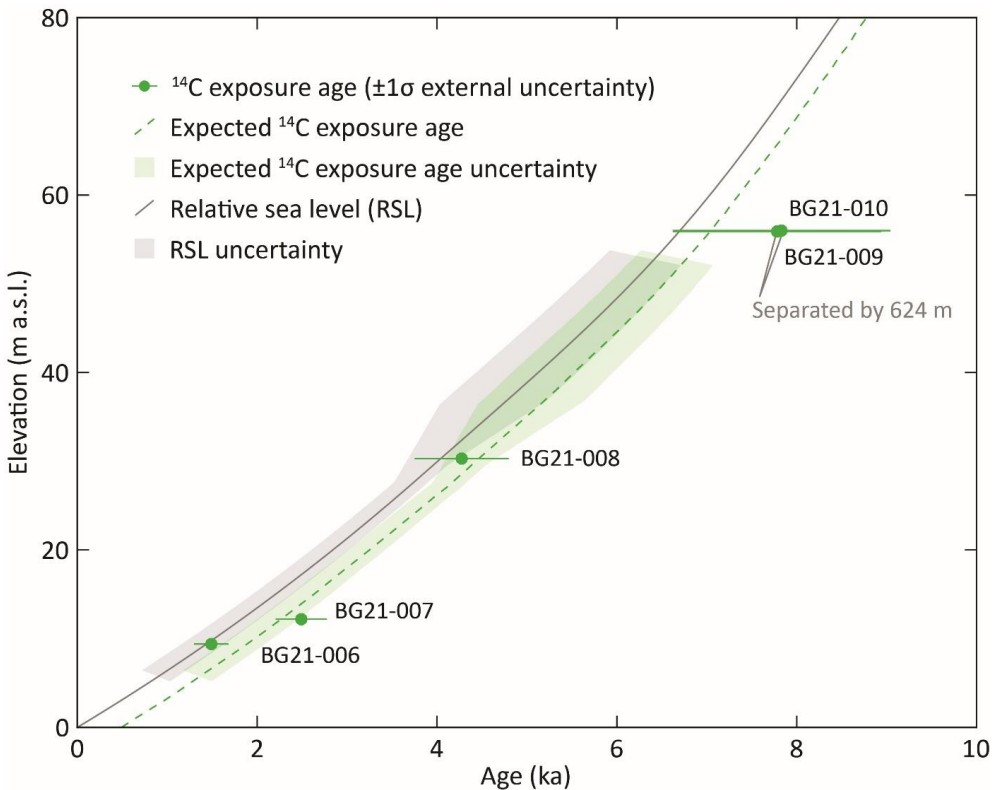

**Figure 3**. Apparent [14]C exposure ages for five Forsmark samples from below the highest shoreline (Fig.
1b; Table 2) with 1σ external uncertainties. The expected exposure ages are calculated assuming the RSL
curve is correct, the [14]C spallation production rate is correct, partial exposure as the sample approaches
the water surface, and full post-glacial exposure for the duration above sea level. Hence, the expected
exposure age curve is a few hundred years older than the RSL curve. The RSL curve is from SKB (2020)
and uncertainties for the 1–6 ka interval are calculated from the original radiocarbon data in Hedenström
and Risberg (2003). The RSL uncertainty envelope is also transposed onto the expected exposure age
curve.



Apparent exposure ages for the five *in situ* $^{14}$C samples located above the highest shoreline in Dalarna
and Gävleborg (Fig. 1a) are shown in Figure 4 and Table 2. The weighted mean age from all five samples
is 11.2 ± 1.3 ka. These data display a $X_R^2$ of 1.78 and a p-value of 0.13 based on 1σ internal uncertainties
(Fig. 4a), which does not support a rejection of the hypothesis that the apparent exposure ages represent
the same population. In addition to the samples being from the same population, the exposure ages are
consistent, within uncertainty, with the expected deglaciation age of 10.8 ± 0.3 ka (Stroeven et al. 2016).
Replicate measurements on sample BG21-002 closely agree and an age based on a weighted mean $^{14}$C
concentration is shown in Figure 4. Sample BG21-001 provides the youngest apparent age but, because
this sample was from a low-profile outcrop (Supplement 1), this age may reflect partial shielding of the
sampled bedrock surface by a past shallow soil cover or perhaps a deeper snow cover than the other
sites. We therefore consider this sample as least likely to provide a reliable age. Removing this sample
from consideration indicates that the remaining four sample sites are more clustered, with an older
weighted mean age of 11.6 ± 1.1 ka, which displays a $X_R^2$ of 0.43 and a p-value of 0.73 based on 1σ
internal uncertainties (Fig. 4b).

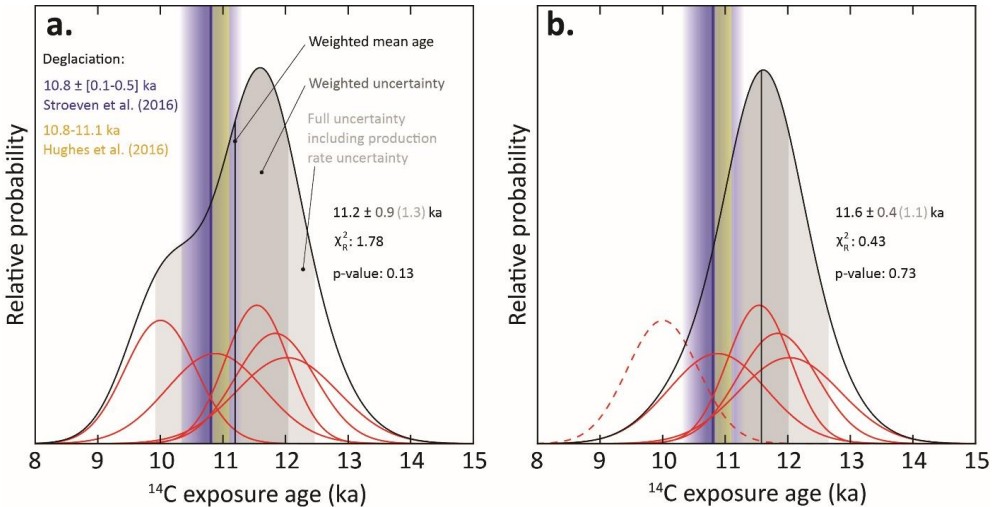

**Figure 4**. Probability density plots of the exposure ages from samples above the highest shoreline (Fig.
1a; Table 2). The individual samples (red curves) display 1σ internal uncertainty (measurement
uncertainty). For the repeat sample BG21-002, the exposure age is calculated with a weighted mean $^{14}$C
concentration using a 2% uncertainty. **(a)** The probability density and data for all five samples. For the
full set of samples, the cosmogenic nuclide ages yield a reduced chi-square ($X_R^2$) of 1.78 and a p-value
of 0.13 based on internal uncertainties, which indicates that they are from the same population. **(b)** The
probability density and data with sample BG21-001 excluded as an outlier. These cosmogenic nuclide
ages yield a $X_R^2$ of 0.43 and a p-value of 0.73 based on internal uncertainties, which again indicate that



they are from the same population. All ages are referenced to the sampling year 2021. The weighted
ages of 11.2 ± 1.3 ka and 11.6 ± 1.1 ka both overlap with the deglaciation age from Stroeven et al. (2016).




Table 1. Extraction and measurement of *in situ* $^{14}C$ at PRIME Lab.

| Sample ID | PCEGS #[a] | PLID[b] | Mass Quartz (g) | C yield (µg) | Diluted mass C (µg) | AMS Split Mass C (µg) | δ$^{13}$C[c] (‰VPDB) | $^{14}$C/$^{13}$C ($10^{-11}$) | $^{14}$C/C$_{total}$ ($10^{-13}$) | $^{14}$C ($10^6$ at) | [$^{14}$C] ($10^5$ at g$^{-1}$)[d] |
|---|---|---|---|---|---|---|---|---|---|---|---|
| BG21-001 | 146 | 202101960 | 5.02378 | 5.0 ± 0.1 | 393.8 ± 4.8 | 382.3 ± 4.6 | -45.9 ± 0.2 | 0.3399 ± 0.0075 | 0.3412 ± 0.0079 | 0.6177 ± 0.0179 | 1.2296 ± 0.0357 |
| BG21-002 | 147 | 202101961 | 5.02383 | 7.8 ± 0.1 | 303.3 ± 3.7 | 294.4 ± 3.6 | -44.8 ± 0.2 | 0.4555 ± 0.0096 | 0.4623 ± 0.0102 | 0.6470 ± 0.0181 | 1.2879 ± 0.0360 |
| BG21-003 | 148 | 202101962 | 5.01070 | 17.6 ± 0.3 | 303.4 ± 3.7 | 294.5 ± 3.6 | -43.9 ± 0.2 | 0.4633 ± 0.0108 | 0.4709 ± 0.0113 | 0.6604 ± 0.0197 | 1.3180 ± 0.0393 |
| BG21-002R | 150 | 202201473 | 5.04116 | 7.7 ± 0.1 | 305.3 ± 3.7 | 296.4 ± 3.6 | -45.2 ± 0.2 | 0.4558 ± 0.0135 | 0.4624 ± 0.0142 | 0.6519 ± 0.0237 | 1.2931 ± 0.0470 |
| BG21-004 | 152 | 202101963 | 5.05927 | 11.9 ± 0.2 | 305.7 ± 3.7 | 296.8 ± 3.6 | -44.6 ± 0.2 | 0.4618 ± 0.0079 | 0.4691 ± 0.0083 | 0.6630 ± 0.0159 | 1.3105 ± 0.0314 |
| BG21-005 | 153 | 202101964 | 5.07578 | 4.6 ± 0.1 | 304.5 ± 3.7 | 295.6 ± 3.6 | -45.4 ± 0.2 | 0.4600 ± 0.0127 | 0.4667 ± 0.0134 | 0.6566 ± 0.0225 | 1.2935 ± 0.0444 |
| BG21-006 | 155 | 202101965 | 5.06572 | 5.5 ± 0.1 | 306.8 ± 3.7 | 297.8 ± 3.6 | -45.2 ± 0.2 | 0.1277 ± 0.0056 | 0.1172 ± 0.0059 | 0.1243 ± 0.0101 | 0.2453 ± 0.0199 |
| BG21-007 | 157 | 202101966 | 5.03589 | 6.9 ± 0.1 | 309.2 ± 3.8 | 300.1 ± 3.7 | -45.0 ± 0.2 | 0.1684 ± 0.0051 | 0.1601 ± 0.0054 | 0.1922 ± 0.0096 | 0.3817 ± 0.0191 |
| BG21-008 | 158 | 202101967 | 5.07653 | 4.0 ± 0.1 | 308.9 ± 3.8 | 299.9 ± 3.6 | -45.4 ± 0.2 | 0.2357 ± 0.0063 | 0.2308 ± 0.0067 | 0.3015 ± 0.0119 | 0.5938 ± 0.0234 |
| BG21-009 | 160 | 202101968 | 5.01906 | 55.3 ± 0.7 | 305.6 ± 3.7 | 296.6 ± 3.6 | -38.0 ± 0.2 | 0.3339 ± 0.0095 | 0.3368 ± 0.0101 | 0.4601 ± 0.0170 | 0.9168 ± 0.0339 |
| BG21-010 | 161 | 202101969 | 4.99961 | 42.2 ± 0.6 | 306.0 ± 3.7 | 297.0 ± 3.6 | -40.1 ± 0.2 | 0.3320 ± 0.0068 | 0.3340 ± 0.0072 | 0.4565 ± 0.0132 | 0.9130 ± 0.0264 |

[a] PCEGS # = sample number in the Purdue Carbon Extraction and Graphitization System
[b] PLID = PRIME Lab ID
[c] Measurement uncertainty of ±0.2 ‰VPDB (where VPDB is Vienna Peedee Belemnite)
[d] Corrected for procedural blank of (5.5952 ± 0.3713) x $10^4$ atoms






Table 2. *In situ* $^{14}$C from quartz, Dalarna-Gävleborg and Forsmark-Uppland.

| Sample ID | Lat (°) | Long (°) | Elevation (m a.s.l.) | Thickness (cm) | Density (g cm$^{-3}$) | Shielding factor | Erosion (cm yr$^{-1}$) | $^{14}$C ± 1σ (10$^2$ at g$^{-1}$) | $^{14}$C Age ± Unc.$^{Int.}$ ± Unc.$^{Ext.}$ [a] (ka) |
|---|---|---|---|---|---|---|---|---|---|
| BG21-001 | 60.47432 | 16.33134 | 236.5 | 3 | 2.7 | 1 | 0 | 1 230 ± 36 | 10.0 ± 1.7 (± 0.6) |
| BG21-002 | 60.40615 | 16.22197 | 212.6 | 3 | 2.7 | 1 | 0 | 1 288 ± 36 | 11.5 ± 2.2 (± 0.7) |
| BG21-002R | 60.40615 | 16.22197 | 212.6 | 3 | 2.7 | 1 | 0 | 1 293 ± 47 | 11.6 ± 2.3 (± 0.9) |
| BG21-003 | 60.38459 | 16.17649 | 216.3 | 3 | 2.7 | 1 | 0 | 1 318 ± 39 | 12.0 ± 2.4 (± 0.8) |
| BG21-004 | 60.38451 | 16.17440 | 217.8 | 3 | 2.7 | 1 | 0 | 1 311 ± 31 | 11.8 ± 2.3 (± 0.6) |
| BG21-005 | 60.36888 | 16.30526 | 248.1 | 3 | 2.7 | 1 | 0 | 1 294 ± 44 | 10.9 ± 2.1 (± 0.8) |
| BG21-006 | 60.38490 | 18.22308 | 9.4 | 3 | 2.7 | 1 | 0 | 245 ± 20 | 1.5 ± 0.2 (± 0.1) |
| BG21-007 | 60.37892 | 18.19129 | 12.2 | 3 | 2.7 | 1 | 0 | 382 ± 19 | 2.5 ± 0.3 (± 0.1) |
| BG21-008 | 60.30504 | 18.04993 | 30.3 | 3 | 2.7 | 1 | 0 | 594 ± 23 | 4.3 ± 0.5 (± 0.2) |
| BG21-009 | 60.22988 | 17.94989 | 56.0 | 3 | 2.7 | 1 | 0 | 917 ± 34 | 7.8 ± 1.2 (± 0.5) |
| BG21-010 | 60.22431 | 17.95051 | 55.9 | 3 | 2.7 | 1 | 0 | 913 ± 26 | 7.8 ± 1.2 (± 0.4) |

[a] Unc.$^{Ext.}$ is external uncertainty and Unc.$^{Int.}$ is internal uncertainty. Both are 1σ.




## 5. Discussion

The *in situ* [14]C bedrock exposure ages from the Forsmark-Uppland transect (i.e., below the highest postglacial shoreline) consistently increase with elevation and overlap the expected exposure age curve, within uncertainty (Fig. 3). Because the apparent exposure ages accurately reflect the timing of landscape emergence, *in situ* [14]C is indicated as having high potential as a chronometer over Late Glacial-Holocene timescales in low relief, low elevation settings. This study adds to precious few demonstrations of the ability of cosmogenic nuclide isotopes to define postglacial landscape emergence above sea level (Briner et al., 2006; Bierman et al., 2018). Briner et al. (2006) present good (visual) congruence with a record of shoreline emergence built from radiocarbon-dated driftwood and fauna by Dyke et al. (1992) using [10]Be measurements on boulders in beaches derived from wave-washed till. Their study also mentions that building a relative sea level curve from pebbles, cobbles and plucked bedrock suffered from inheritance problems, an experience shared by Matmon et al. (2003) while attempting the dating of chert on beach ridges in southern Israel and heeded by Bierman et al. (2018). Bierman et al. (2018) successfully dated landscape emergence on Greenland using [10]Be across a range of settings, including bedrock below the highest shoreline, cobbles from beach ridges at the highest shoreline, and boulders and bedrock above the highest shoreline. They note that success hinges on the requirement of warm-based ice and deep glacial erosion in exposing bedrock devoid of an inherited cosmogenic nuclide inventory. In many regions, however, including east-central Sweden and more widely in Fennoscandia, these requirements are not met either because of cold-based conditions (Patton et al., 2016; Stroeven et al., 2016) or weakly erosive warm-based ice such as at Forsmark (Hall et al., 2019; SKB, 2020), during all or much of glacial time. Cosmogenic nuclide inheritance is therefore a part of the landscape fabric. Bierman et al. (2018) advocate the use of *in situ* [14]C as a methodology to circumvent inheritance problems. Our study is the first to follow-up on that suggestion, and shows, convincingly, that using *in situ* [14]C can extend the study of landscape rebound to regions where ice sheet erosion was insufficiently deep to allow for the application of long-lived nuclides.

Five bedrock samples from above the highest postglacial shoreline are well-clustered and the weighted mean age (and full uncertainty) of 11.2 ± 1.3 ka overlaps with the predicted deglaciation age of 10.8 ± 0.3 ka (Fig. 4a; Hughes et al., 2016; Stroeven et al., 2016). Removing the youngest age from consideration results in more strongly clustered ages (Fig. 4b) and an older mean weighted age of 11.6 ± 1.1 ka, which still overlaps the predicted deglaciation age, within uncertainty. We therefore do not further discriminate between these results. Because derived exposure ages overlap with the predicted deglaciation age, we further infer that the *in situ* [14]C samples, including those located below the highest postglacial shoreline, within uncertainty, lack inheritance from previous exposure. This implies that the last ice sheet advanced over the study area soon after 35 ka, in accordance with previous inferences



for Forsmark (SKB, 2020). An alternative interpretation is that the last ice sheet advanced more recently
but that glacial erosion during MIS2 was sufficiently deep to remove any nuclide inheritance.
Our *in situ* [14]C data from above the highest (postglacial) shoreline demonstrate good potential for this
nuclide to help constrain the deglaciation chronology of former ice sheets. This is especially true for
regions with thin drift, abundant bedrock exposures, and lacking moraines outlining successive retreat
stages. In Fennoscandia, thin drift conditions occur commonly (cf. Kleman et al., 2008) and ice sheet
retreat appears to have proceeded uninterrupted inside the Younger Dryas moraine belt (apart from
the Central Finland Ice-Marginal Formation; e.g., Rainio et al., 1986; Stroeven et al., 2016). Whereas
the post-Younger Dryas deglaciation of east-central Sweden is well constrained by clay-varve
chronology (Strömberg, 1989) below the highest postglacial shoreline, there are vast areas above the
highest shoreline that remain poorly constrained by data (Stroeven et al. 2016). In addition to a lack of
datable deglacial landforms, this is attributable to glacial erosion of bedrock having frequently been
insufficient to remove inventories of long half-life [10]Be and [26]Al (Patton et al., 2022), thereby leaving
nuclides inherited from exposure prior to the last glaciation (Heyman et al., 2011; Stroeven et al., 2016).
Because of the short [14]C half-life and an improved sampling methodology, *in situ* [14]C may now be a
prime candidate nuclide to be included in last deglaciation studies on glaciated cratons, such as the
dating of boulders deposited along glacial flowlines; a technique practiced successfully using [10]Be
(Margold et al., 2019; Norris et al., 2022).

**6. Conclusion**
Ten *in situ* [14]C measurements on bedrock are consistent with a RSL curve for Forsmark derived from
organic radiocarbon dating of basal sediments in isolation basins and the Fennoscandian Ice Sheet
deglaciation chronologies from Stroeven et al. (2016) and Hughes et al. (2016). This study introduces
the use of *in situ* [14]C in Fennoscandian Ice Sheet paleoglaciology and outlines a promise of its use as a
basis for supporting future shoreline displacement studies and for tracking the deglaciation in areas
that lack datable organic material and where [10]Be and [26]Al routinely return complex exposure results.

**Data availability.** Data are available in Supplements 1-3. LiDAR data used in the study can
downloaded from https://www.lantmateriet.se
**Author contributions.** BWG and APS initiated the study, with support from KH and JON, and drafted
the manuscript. BWG, APS, and AL did the sampling. AL did petrological analyses of the sampled
bedrock. NAL completed sample preparation for AMS and provided the results. JH carried out
cosmogenic nuclide production rate and exposure age calculations. MWC oversaw the AMS. All
authors revised the manuscript.



**Competing interests.** The contact author has declared that none of the authors has any competing interests.

**Disclaimer.** Publisher's note: Copernicus Publications remains neutral with regard to jurisdictional claims in published maps and institutional affiliations.

**Acknowledgements.** We thank Johan Liakka (SKB) for his support in completing this study.

**Financial support.** This research was supported by the Swedish Nuclear Fuel and Waste Management Company.

Review statement.

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
