# Peer review of "using in situ cosmogenic 14C from quartz 2 3 4 Bradley W. Goodfellow1\* 5 Arjen P. Stroeven2,3 6 Nathaniel A. Lifton4,5 7 Jakob Heyman6 8 Alexander Lewerentz1 Kristina Hippe7 9 10 Jens-Ove Näslund8 11 Marc W. Caffee4,5 12 1"

_EGUsphere, 2023_

## Author Comment (AC1)

**Replies to reviewer 2, Nicolás Young**

This manuscript presents a series of in situ 14C measurements in bedrock surfaces with the goal of assessing their reliability in 1) constraining the rate/timing of landscape emergence, and 2) constraining the timing of local deglaciation. The manuscript is set up by pointing out that the author's field area already has a pretty well constrained emergence curve and an independent constraint on the timing of local deglaciation (some mixture of traditional 14C and 10Be from Stroeven et al., 2016). Thus, the author's hope that their in situ 14C measurements "match" the independent controls as a test of their usefulness in this type of glacial setting. The approach here is to present a new tool to the community and show how it might work/behave in an ideal setting. By and large the in situ 14C measurements here match the independent record….things appear to have "worked."

I really enjoyed reading this manuscript. I think the authors set up the study nicely, gave the appropriate amount of background information, kept it shirt and focused, and didn't try to turn this dataset/manuscript into something it's not. Well done. Dataset and measurements look solid. This manuscript will certainly be of interest to the cosmogenic nuclide community, in particular users of in situ 14C. I think this manuscript is perfect for Ghron.

I do not have many suggestions here; again, this manuscript is already a nice little package that presents a clean story. I have a few minor comments, and then one more major comment. The latter is (likely) more directed at Nat and is the product of a timely independent conversation I have been having with Nat over the last week. The authors should pay particular attention to this and make sure it gets fixed not only for this manuscript but in relevant databases. Even though this is a more major comment, it should not affect the overall conclusions of this manuscript here.

**Thanks for your positive and helpful feedback on our manuscript.**

Major comment:

Authors did a great job at explaining how in situ 14C ages were calculated and also provide a great figure (Fig. 2) that shows the 14C calibration datasets that go into calculating the spallation 14C production rate of "13.35+-1.13" atoms/g/yr. Glancing at panel A in Figure 2 I thought that the New Zealand and Greenland calibration numbers looked a little high relative to the other calibration datasets. These calibration 14C concentrations, in addition to all the others, were recalculated using the methods of Hippe and Lifton, 2014. For the most part, this results in a very small adjustment to the 14C concentrations. However, for the New Zealand measurements the change is significant larger. The recalculated concentrations are ~6-16% higher. This was brought to my attention last week and we have had an email chain going on trying to figure out what was happening. It appears that this issue has been identified – the primary culprit was that in their original publications only the Fraction Modern was reported *instead* of the 13C-corrected Fraction Modern, so when the un-corrected Fraction Modern values were used in recalculation using the Hippe and Lifton, 2014 methods, its resulting in significantly higher sample 14C concentrations.

I looked through the code that the authors use here for calculating in situ 14C ages, and I couldn't find the individual calibration numbers, I did see the "13.35+-1.13" atoms/g/yr" in the constants file. But I searched around on the calculator website that the authors reference and

found this file, which I believe is what the authors are using to derive the global in situ 14C production rate: https://expage.github.io/data/prodrate/P-202306-input.txt I scrolled down to the 14C concentrations and, yes, what is here are the significantly higher New Zealand and Greenland concentrations. I also think these are the concentrations posted in the ICE-D database.

I raise this issue because those two datasets comprise ~15% of all the calibration measurements. With the pending concentration changes, the global production rate used here is going to get lower, and the uncertainty will likely improve. This will change the absolute in situ 14C ages in this manuscript a bit, but I believe with the error bars, all the in situ 14C ages will still overlap with the independent constraints (e.g., Fig. 3). The calibration concentrations need to be updated here, including in the relevant databases, and then the absolute ages need to be updated.

**Thanks for identifying this important issue, which we have now remedied. Co-author Lifton corresponded with Drs. Young and Schimmelpfennig (first authors of the manuscripts detailing the Greenland and New Zealand calibration datasets, respectively) and identified the main issues responsible for the discrepancy – 1) the published Fm values were not corrected for stable C isotopic composition, and 2) the more significant correction for the mass-dependent graphitization blank was not included in the published Fm values, nor was the formula used to make such corrections published. We have now calculated the corrected concentrations for those two calibration datasets, and have recalculated the global production rate to 12.81 at g$^{-1}$ yr$^{-1}$ – a ca. 4% decrease from 13.35 at g$^{-1}$ yr$^{-1}$. Updated concentrations for those two calibration sites are now also updated in the ICE-D Calibration database, and a corrigendum is in preparation for the Koester and Lifton (2023) citation from which the original recalculated data were taken.**

Minor comments:

Lines 80-81 – what about marine terraces? I think that's a common term/setting for developing emergence curves.

**There are no marine terraces here. Terrace formation is inhibited because the landscape is formed on tough, glacially scoured, crystalline bedrock (with only a patchy till cover), it's a highly fetch-limited environment, which limits wave energy necessary to terrace formation, and rapid isostatic rebound may have led to insufficient time for a terrace to form at a particular relative sea level.**

Lines 142-144 – wondering how applicable this 200m marine limit located 100 km away is to your site? Are there any closer marine limit benchmarks? Doesn't really change anything, just curious.

**We have located sites above the highest postglacial shoreline that are as close as possible to our Forsmark sites that were all located below sea level following deglaciation of the last ice sheet. Because the landscape is as low lying and low relief as it is, the nearest sites are about 100 km west. Once we found locations that were above the highest postglacial shoreline, we then had to locate outcrops in bedrock suitable for *in situ* [14]C analyses, which has some additional minor influence on distances from Forsmark.**

Lines 225-235/Figure 2 – see main comment above

**Noted!**

Lines 398-403 – Agree with everything here. Not saying it's required, but if the authors wanted, it would be pretty simple to model a few endmember scenarios of MIS 3 exposure/MIS 2 cover +erosion histories to give the reader a sense on what it would take to arrive at the measured concentrations. For example, what would the evolution of 14C concentrations look like if you dosed the surface for 10 kyr during MIS, cover it during MIS 2 with no erosion, then re-expose at ~10 ka? Maybe just a few of these using *reasonable* exposure/burial/erosion rate constraints to give the reader how in situ14C concentrations may have evolved through the latter half of the glacial cycle?

**Thanks for this good suggestion. We have added a new figure, "Figure 5", that shows the hypothetical development of $^{14}$C concentration in the five samples above the highest shoreline in two different ice cover scenarios. We assume no glacial or interglacial erosion, continuous exposure during ice-free periods, and full shielding from cosmic rays during periods with ice cover. The simulations are done over the last 80 ka with ice cover during the periods 70-57 and 35-10.7 ka BP for the longer ice cover scenario and 66-60 and 28-10.7 ka BP for the short ice cover scenario. The simulations clearly show that if the MIS-2 ice advanced over the samples at 28 ka or earlier, the pre-MIS-2 ice history does not matter much for the present-day $^{14}$C concentration.**

Replying to my own review here, just want to be clear. The main issue is that the 13C **+ Blank** corrected Fraction Modern value needed to be used for the 14C calibration samples I mentioned. Sorry for the confusion.

**Thank you again for clarifying this, we have adjusted the data and figures accordingly – see response above.**

---

## Author Comment (AC3)

**Replies to reviewer 1**

This paper reports in situ 14C data from Sweden and aims to assess the accuracy of this dating method by comparing it to a relative sea level (RSL) curve based on radiocarbon dating of organic material in isolated basins and a local deglaciation timing determined from a clay varve chronology. The authors collected samples of granitoid bedrock both below and above the highest postglacial shoreline and found that the in situ 14C measurements provided reliable age constraints, closely aligning with the RSL curve and local deglaciation chronology, demonstrating its utility for accurately dating ice sheet deglaciation and postglacial exposure in regions where other methods yield complex results.

It is a short and concise paper and I only have some comments to address:

1. Please include the statement "not affected by the marine reservoir effect" in line 92.

   **We disagree. The comment makes no sense.**

2. In lines 133-137, I kindly request a more in-depth discussion of the studies that this paper references, as the discussion and conclusion rely heavily on these two papers, focusing on the reliability of the quoted ages.

   **We have expanded the review of these studies to (l. 135-143): "*The Hughes et al. (2016) reconstruction relies primarily upon chronological constraints supplied from radiocarbon, thermal luminescence, optically stimulated luminescence (OSL), infrared stimulated luminescence, electron spin resonance, terrestrial cosmogenic nuclide (TCN), and U series dating. Published landform data, mostly with respect to end moraines and generally accepted correlations of ice-margin positions between individual moraines, provide complementary evidence. In contrast, the Stroeven et al. (2016) reconstruction combines geomorphological constraints for ice sheet margin outlines, including ice-marginal depositional landforms and meltwater channels, ice-dammed lakes, eskers, lineations, and striae, with chronological constraints supplied by radiocarbon, varve, OSL, and TCN dating.*"**

3. The blue sign on the Figure map, indicating 'below the highest shoreline,' is confusing, considering that the Dalarna region pertains to the area above the highest shoreline. I recommend its removal, as it does not contribute to a better understanding of the research.

   **Most of this area is below the highest postglacial shoreline. We sampled sites located on what were islands upon deglaciation, as we have illustrated in the figure panel, which we think should remain as is. To provide further clarity we have amended the caption of Figure 1 to: "*The five Dalarna-Gävleborg sample sites are located on what were islands above the highest postglacial shoreline*".**

4. Given that two of the authors have contributed to the in-situ C-14 calculation paper published in Radiocarbon 2014, it would be beneficial to incorporate the VTS value into the analysis. Additionally, please specify whether OX-I or Ox-II was used for data reduction. If any dilution correction was applied, ensure that it is included in the table. Furthermore, I kindly request comprehensive data for the blank value, including gas

yield. It would also be greatly appreciated if you could include relevant information on Cronus A or another intercomparison sample closely aligned with the samples presented in Table 1. Based on the given data, I calculate for BG21-001 1.28-1.32*105 atoms/g but I'm unsure why the AMS split is less than the sample+dilution. Is this due to stable isotope fractionation or transfer loss?

**We thank the reviewer for this comment, which has led to several improvements (see below). However, we are not sure what a VTS value is (not listed in Hippe and Lifton, 2014, as implied) so we cannot respond more usefully here. If it refers to the $CO_2$ volume, that can be converted quite straightforwardly to the equivalent mass of C (which we present), and would be redundant in our view. OX-2 is the measurement standard used (but that standard is referenced to OX-1). We are including a note in the tables. The interested reader can find representative CRONUS-A values in Lifton et al. (2023). Diluted sample mass is the correct mass to use for concentration determination (total of C yield + added $^{14}C$-free $CO_2$), as that is reflected in the measured $^{14}C/^{13}C$ ratio. AMS split mass is diluted sample mass less a small aliquot (typically ca. 9 μg C) for offline stable carbon isotopic measurement. The AMS split mass is used for the mass-dependent graphitization blank correction (see Lifton et al., 2023, for example). Notes clarifying this have been added to the table below (Table 1 in the manuscript). Relevant procedural blank data has now been tabulated and included at the same level of detail as the samples, as suggested.**

5. It is important to include a sentence discussing the blank effect, especially for samples BG21-006, 007, and 008. Please elaborate on the implications for ages if the blank were 5000-10000 atoms higher or lower.

**We are unsure of the 'blank effect' to which the reviewer is referring. The values in the total $^{14}C$ inventory column already reflect the subtracted procedural blank. We have clarified this in the table notes - any shifts in the blank would have only a small effect on the total remaining inventory. Yes, the subtraction is ca. 15-30% of the total measured values of the samples listed, but as one can see from the now-tabulated blank data (appended to Table 1, below), the blank is well constrained during this period, as represented by the mean and standard deviation that are used. In our opinion, it is uninformative to speculate about whether the mean is higher or lower than what has been demonstrably stable during the period spanning the sample analyses, when the variability is well-quantified by the mean and standard deviation. If we restrict the blanks to those immediately bracketing the Forsmark samples (PCEGS-145 and PCEGS-163), the resulting change in the mean is less than 3000 atoms (out of >10$^5$) between the full mean and the mean of just the bracketing values, and well within 1σ standard deviation of the broader mean blank. So, not much implication to ages at all – only 600 atoms/g change in concentration. We have therefore added text clarifying this to the discussion section (lines 272-274). "Analytical results for *in situ* $^{14}C$ samples and procedural blanks are presented in Table 1. The mean and standard deviation are used to correct measured $^{14}C$ sample inventories (Table 1) because procedural blanks are well-constrained during the analytical time frame."**

6. My knowledge of MATLAB does not allow me from checking the script attached to the paper, but the currently published years appear significantly smaller when recalculated

with the online exposure age calculator v3. Please address this discrepancy in line 235 and provide reasons for it.

**The reason for the online exposure age calculator v3 yielding clearly older exposure ages is that the v3 calculator uses a lower default in situ $^{14}$C production rate. This text about the $^{14}$C production rate is from the v3 calculator documentation:**

**"$^{14}$C is calibrated from some measurements of the CRONUS-A sample (saturated) by Brent Goehring in the now-defunct Tulane lab. This needs work. It is also not integrated with the ICE-D database. At present, I recommend supplying your own calibration for $^{14}$C calculations."**

**When we use the same production rate calibration dataset as used for the expage-202403 calculator production rate, we get similar ages from the v3 calculator.**

7. I do not understand the reason for excluding the first sample if it passed the Chi-square test. Please provide a stronger explanation for this decision.

**We now include all data in a single panel figure to remove speculation.**

**Table 1 - *In situ* [14]C sample measurement details**

| SAMPLE | PCEGS[1] # | PLID[2] | Mass Quartz (g) | C yield (ug) | Diluted Mass C (ug) | AMS Split Mass C[3] (ug) | $\delta^{13}C$ (‰ VPDB) | $^{14}C/^{13}C$[4] ($10^{-12}$) | $^{14}C/C_{total}$[5] ($10^{-14}$) | $^{14}C$[6] ($10^5$ at) | $[^{14}C]$ ($10^5$ at g$^{-1}$) |
|---|---|---|---|---|---|---|---|---|---|---|---|
| BG21-001 | PCEGS-146 | 202101960 | 5.02378 | 5.0 ± 0.1 | 393.8 ± 4.8 | 382.3 ± 4.6 | -45.9 ± 0.2 | 3.3992 ± 0.0745 | 3.4118 ± 0.0785 | 6.1771 ± 0.1793 | 1.2296 ± 0.0357 |
| BG21-002 | PCEGS-147 | 202101961 | 5.02383 | 7.8 ± 0.1 | 303.3 ± 3.7 | 294.4 ± 3.6 | -44.8 ± 0.2 | 4.5548 ± 0.0964 | 4.6226 ± 0.1016 | 6.4703 ± 0.1806 | 1.2879 ± 0.0360 |
| BG21-003 | PCEGS-148 | 202101962 | 5.01070 | 17.6 ± 0.3 | 303.4 ± 3.7 | 294.5 ± 3.6 | -43.9 ± 0.2 | 4.6325 ± 0.1075 | 4.7091 ± 0.1134 | 6.6042 ± 0.1969 | 1.3180 ± 0.0393 |
| BG21-002R | PCEGS-150 | 202201473 | 5.04116 | 7.7 ± 0.1 | 305.3 ± 3.7 | 296.4 ± 3.6 | -45.2 ± 0.2 | 4.5575 ± 0.1350 | 4.6239 ± 0.1422 | 6.5186 ± 0.2368 | 1.2931 ± 0.0470 |
| BG21-004 | PCEGS-152 | 202101963 | 5.05927 | 11.9 ± 0.2 | 305.7 ± 3.7 | 296.8 ± 3.6 | -44.6 ± 0.2 | 4.6181 ± 0.0789 | 4.6905 ± 0.0832 | 6.6300 ± 0.1588 | 1.3105 ± 0.0314 |
| BG21-005 | PCEGS-153 | 202101964 | 5.07578 | 4.6 ± 0.1 | 304.5 ± 3.7 | 295.6 ± 3.6 | -45.4 ± 0.2 | 4.5997 ± 0.1272 | 4.6668 ± 0.1339 | 6.5656 ± 0.2251 | 1.2935 ± 0.0444 |
| BG21-006 | PCEGS-155 | 202101965 | 5.06572 | 5.5 ± 0.1 | 306.8 ± 3.7 | 297.8 ± 3.6 | -45.2 ± 0.2 | 1.2766 ± 0.0562 | 1.1715 ± 0.0594 | 1.2426 ± 0.1010 | 0.2453 ± 0.0199 |
| BG21-007 | PCEGS-157 | 202101966 | 5.03589 | 6.9 ± 0.1 | 309.2 ± 3.8 | 300.1 ± 3.7 | -45.0 ± 0.2 | 1.6838 ± 0.0507 | 1.6007 ± 0.0536 | 1.9221 ± 0.0960 | 0.3817 ± 0.0191 |
| BG21-008 | PCEGS-158 | 202101967 | 5.07653 | 4.0 ± 0.1 | 308.9 ± 3.8 | 299.9 ± 3.6 | -45.4 ± 0.2 | 2.3565 ± 0.0634 | 2.3076 ± 0.0669 | 3.0145 ± 0.1185 | 0.5938 ± 0.0234 |
| BG21-009 | PCEGS-160 | 202101968 | 5.01906 | 55.3 ± 0.7 | 305.6 ± 3.7 | 296.6 ± 3.6 | -38.0 ± 0.2 | 3.3393 ± 0.0946 | 3.3681 ± 0.1005 | 4.6013 ± 0.1703 | 0.9168 ± 0.0339 |
| BG21-010 | PCEGS-161 | 202101969 | 4.99961 | 42.2 ± 0.6 | 306.0 ± 3.7 | 297.0 ± 3.6 | -40.1 ± 0.2 | 3.3197 ± 0.0680 | 3.3399 ± 0.0721 | 4.5648 ± 0.1321 | 0.9130 ± 0.0264 |
| **Procedural Blanks** | | | | | | | | | | | |
| PB2-03222022 | PCEGS-135 | 202201450 | -- | 1.4 ± 0.1 | 305.2 ± 3.7 | 296.2 ± 3.6 | -40.2 ± 0.2 | 0.4853 ± 0.0298 | 0.3413 ± 0.0320 | 0.5222 ± 0.0493 | -- |
| PB2-04212022 | PCEGS-145 | 202201452 | -- | 1.8 ± 0.1 | 307.0 ± 3.7 | 298.0 ± 3.6 | -46.0 ± 0.2 | 0.5182 ± 0.0273 | 0.3731 ± 0.0292 | 0.5742 ± 0.0455 | -- |
| PB2-05212022 | PCEGS-163 | 202201454 | -- | 2.3 ± 0.1 | 307.4 ± 3.7 | 298.4 ± 3.6 | -46.0 ± 0.2 | 0.5364 ± 0.0315 | 0.3922 ± 0.0335 | 0.6045 ± 0.0521 | -- |
| PB2-06022022 | PCEGS-169 | 202201459 | -- | 2.3 ± 0.1 | 307.3 ± 3.7 | 298.3 ± 3.6 | -40.3 ± 0.2 | 0.4920 ± 0.0291 | 0.3486 ± 0.0312 | 0.5371 ± 0.0486 | -- |

*Mean ± 1σ (All blanks)*  *0.5595 ± 0.0371*

*Mean ± 1σ (145,163 only)*  0.5894 ± 0.0214

Notes

1    Purdue Carbon Extraction and Graphitization System
2    Prime Lab ID
3    Mass graphitized for AMS analysis after small aliquot (ca. 9 ug C) taken for stable C isotopic analysis offline
4    Measured relative to OX-2 standard
5    Corrected for mass-dependent graphitization blank (based on AMS Split Mass C) and stable C composition
6    Sample values calculated using Diluted Mass C and corrected for mean procedural blank (All blanks)

---

## Author Response (AR2)

**Replies to reviewer 1**

This paper reports in situ 14C data from Sweden and aims to assess the accuracy of this dating method by comparing it to a relative sea level (RSL) curve based on radiocarbon dating of organic material in isolated basins and a local deglaciation timing determined from a clay varve chronology. The authors collected samples of granitoid bedrock both below and above the highest postglacial shoreline and found that the in situ 14C measurements provided reliable age constraints, closely aligning with the RSL curve and local deglaciation chronology, demonstrating its utility for accurately dating ice sheet deglaciation and postglacial exposure in regions where other methods yield complex results.

It is a short and concise paper and I only have some comments to address:

1.  Please include the statement "not affected by the marine reservoir effect" in line 92.

    **We disagree. The comment makes no sense.**

2.  In lines 133-137, I kindly request a more in-depth discussion of the studies that this paper references, as the discussion and conclusion rely heavily on these two papers, focusing on the reliability of the quoted ages.

    **We have expanded the review of these studies to (l. 135-143): "*The Hughes et al. (2016) reconstruction relies primarily upon chronological constraints supplied from radiocarbon, thermal luminescence, optically stimulated luminescence (OSL), infrared stimulated luminescence, electron spin resonance, terrestrial cosmogenic nuclide (TCN), and U series dating. Published landform data, mostly with respect to end moraines and generally accepted correlations of ice-margin positions between individual moraines, provide complementary evidence. In contrast, the Stroeven et al. (2016) reconstruction combines geomorphological constraints for ice sheet margin outlines, including ice-marginal depositional landforms and meltwater channels, ice-dammed lakes, eskers, lineations, and striae, with chronological constraints supplied by radiocarbon, varve, OSL, and TCN dating.*"**

3.  The blue sign on the Figure map, indicating 'below the highest shoreline,' is confusing, considering that the Dalarna region pertains to the area above the highest shoreline. I recommend its removal, as it does not contribute to a better understanding of the research.

    **Most of this area is below the highest postglacial shoreline. We sampled sites located on what were islands upon deglaciation, as we have illustrated in the figure panel, which we think should remain as is. To provide further clarity we have amended the caption of Figure 1 to: "*The five Dalarna-Gävleborg sample sites are located on what were islands above the highest postglacial shoreline*".**

4.  Given that two of the authors have contributed to the in-situ C-14 calculation paper published in Radiocarbon 2014, it would be beneficial to incorporate the VTS value into the analysis. Additionally, please specify whether OX-I or Ox-II was used for data reduction. If any dilution correction was applied, ensure that it is included in the table. Furthermore, I kindly request comprehensive data for the blank value, including gas

yield. It would also be greatly appreciated if you could include relevant information on Cronus A or another intercomparison sample closely aligned with the samples presented in Table 1. Based on the given data, I calculate for BG21-001 $1.28-1.32*10^5$ atoms/g but I'm unsure why the AMS split is less than the sample+dilution. Is this due to stable isotope fractionation or transfer loss?

**We thank the reviewer for this comment, which has led to several improvements (see below). However, we are not sure what a VTS value is (not listed in Hippe and Lifton, 2014, as implied) so we cannot respond more usefully here. If it refers to the $CO_2$ volume, that can be converted quite straightforwardly to the equivalent mass of C (which we present), and would be redundant in our view. OX-2 is the measurement standard used (but that standard is referenced to OX-1). We are including a note in the tables. The interested reader can find representative CRONUS-A values in Lifton et al. (2023). Diluted sample mass is the correct mass to use for concentration determination (total of C yield + added $^{14}$C-free $CO_2$), as that is reflected in the measured $^{14}$C/$^{13}$C ratio. AMS split mass is diluted sample mass less a small aliquot (typically ca. 9 µg C) for offline stable carbon isotopic measurement. The AMS split mass is used for the mass-dependent graphitization blank correction (see Lifton et al., 2023, for example). Notes clarifying this have been added to the table below (Table 1 in the manuscript). Relevant procedural blank data has now been tabulated and included at the same level of detail as the samples, as suggested.**

5. It is important to include a sentence discussing the blank effect, especially for samples BG21-006, 007, and 008. Please elaborate on the implications for ages if the blank were 5000-10000 atoms higher or lower.

   **We are unsure of the 'blank effect' to which the reviewer is referring. The values in the total $^{14}$C inventory column already reflect the subtracted procedural blank. We have clarified this in the table notes - any shifts in the blank would have only a small effect on the total remaining inventory. Yes, the subtraction is ca. 15-30% of the total measured values of the samples listed, but as one can see from the now-tabulated blank data (appended to Table 1, below), the blank is well constrained during this period, as represented by the mean and standard deviation that are used. In our opinion, it is uninformative to speculate about whether the mean is higher or lower than what has been demonstrably stable during the period spanning the sample analyses, when the variability is well-quantified by the mean and standard deviation. If we restrict the blanks to those immediately bracketing the Forsmark samples (PCEGS-145 and PCEGS-163), the resulting change in the mean is less than 3000 atoms (out of >$10^5$) between the full mean and the mean of just the bracketing values, and well within 1σ standard deviation of the broader mean blank. So, not much implication to ages at all – only 600 atoms/g change in concentration. We have therefore added text clarifying this to the discussion section (lines 272-274). "Analytical results for _in situ_ $^{14}$C samples and procedural blanks are presented in Table 1. The mean and standard deviation are used to correct measured $^{14}$C sample inventories (Table 1) because procedural blanks are well-constrained during the analytical time frame."**

6. My knowledge of MATLAB does not allow me from checking the script attached to the paper, but the currently published years appear significantly smaller when recalculated

with the online exposure age calculator v3. Please address this discrepancy in line 235 and provide reasons for it.

**The reason for the online exposure age calculator v3 yielding clearly older exposure ages is that the v3 calculator uses a lower default in situ $^{14}$C production rate. This text about the $^{14}$C production rate is from the v3 calculator documentation:**

**"$^{14}$C is calibrated from some measurements of the CRONUS-A sample (saturated) by Brent Goehring in the now-defunct Tulane lab. This needs work. It is also not integrated with the ICE-D database. At present, I recommend supplying your own calibration for $^{14}$C calculations."**

**When we use the same production rate calibration dataset as used for the expage-202403 calculator production rate, we get similar ages from the v3 calculator.**

7. I do not understand the reason for excluding the first sample if it passed the Chi-square test. Please provide a stronger explanation for this decision.

    **We now include all data in a single panel figure to remove speculation.**

**Table 1 - *In situ* [14]C sample measurement details**

| SAMPLE | PCEGS[1] # | PLID[2] | Mass Quartz (g) | C yield (ug) | Diluted Mass C (ug) | AMS Split Mass C[3] (ug) | $\delta^{13}C$ (‰ VPDB) | $^{14}C/^{13}C$[4] ($10^{-12}$) | $^{14}C/C_{total}$[5] ($10^{-14}$) | $^{14}C$[6] ($10^5$ at) | $[^{14}C]$ ($10^5$ at g$^{-1}$) |
|---|---|---|---|---|---|---|---|---|---|---|---|
| BG21-001 | PCEGS-146 | 202101960 | 5.02378 | 5.0 ± 0.1 | 393.8 ± 4.8 | 382.3 ± 4.6 | -45.9 ± 0.2 | 3.3992 ± 0.0745 | 3.4118 ± 0.0785 | 6.1771 ± 0.1793 | 1.2296 ± 0.0357 |
| BG21-002 | PCEGS-147 | 202101961 | 5.02383 | 7.8 ± 0.1 | 303.3 ± 3.7 | 294.4 ± 3.6 | -44.8 ± 0.2 | 4.5548 ± 0.0964 | 4.6226 ± 0.1016 | 6.4703 ± 0.1806 | 1.2879 ± 0.0360 |
| BG21-003 | PCEGS-148 | 202101962 | 5.01070 | 17.6 ± 0.3 | 303.4 ± 3.7 | 294.5 ± 3.6 | -43.9 ± 0.2 | 4.6325 ± 0.1075 | 4.7091 ± 0.1134 | 6.6042 ± 0.1969 | 1.3180 ± 0.0393 |
| BG21-002R | PCEGS-150 | 202201473 | 5.04116 | 7.7 ± 0.1 | 305.3 ± 3.7 | 296.4 ± 3.6 | -45.2 ± 0.2 | 4.5575 ± 0.1350 | 4.6239 ± 0.1422 | 6.5186 ± 0.2368 | 1.2931 ± 0.0470 |
| BG21-004 | PCEGS-152 | 202101963 | 5.05927 | 11.9 ± 0.2 | 305.7 ± 3.7 | 296.8 ± 3.6 | -44.6 ± 0.2 | 4.6181 ± 0.0789 | 4.6905 ± 0.0832 | 6.6300 ± 0.1588 | 1.3105 ± 0.0314 |
| BG21-005 | PCEGS-153 | 202101964 | 5.07578 | 4.6 ± 0.1 | 304.5 ± 3.7 | 295.6 ± 3.6 | -45.4 ± 0.2 | 4.5997 ± 0.1272 | 4.6668 ± 0.1339 | 6.5656 ± 0.2251 | 1.2935 ± 0.0444 |
| BG21-006 | PCEGS-155 | 202101965 | 5.06572 | 5.5 ± 0.1 | 306.8 ± 3.7 | 297.8 ± 3.6 | -45.2 ± 0.2 | 1.2766 ± 0.0562 | 1.1715 ± 0.0594 | 1.2426 ± 0.1010 | 0.2453 ± 0.0199 |
| BG21-007 | PCEGS-157 | 202101966 | 5.03589 | 6.9 ± 0.1 | 309.2 ± 3.8 | 300.1 ± 3.7 | -45.0 ± 0.2 | 1.6838 ± 0.0507 | 1.6007 ± 0.0536 | 1.9221 ± 0.0960 | 0.3817 ± 0.0191 |
| BG21-008 | PCEGS-158 | 202101967 | 5.07653 | 4.0 ± 0.1 | 308.9 ± 3.8 | 299.9 ± 3.6 | -45.4 ± 0.2 | 2.3565 ± 0.0634 | 2.3076 ± 0.0669 | 3.0145 ± 0.1185 | 0.5938 ± 0.0234 |
| BG21-009 | PCEGS-160 | 202101968 | 5.01906 | 55.3 ± 0.7 | 305.6 ± 3.7 | 296.6 ± 3.6 | -38.0 ± 0.2 | 3.3393 ± 0.0946 | 3.3681 ± 0.1005 | 4.6013 ± 0.1703 | 0.9168 ± 0.0339 |
| BG21-010 | PCEGS-161 | 202101969 | 4.99961 | 42.2 ± 0.6 | 306.0 ± 3.7 | 297.0 ± 3.6 | -40.1 ± 0.2 | 3.3197 ± 0.0680 | 3.3399 ± 0.0721 | 4.5648 ± 0.1321 | 0.9130 ± 0.0264 |
| **Procedural Blanks** | | | | | | | | | | | |
| PB2-03222022 | PCEGS-135 | 202201450 | -- | 1.4 ± 0.1 | 305.2 ± 3.7 | 296.2 ± 3.6 | -40.2 ± 0.2 | 0.4853 ± 0.0298 | 0.3413 ± 0.0320 | 0.5222 ± 0.0493 | -- |
| PB2-04212022 | PCEGS-145 | 202201452 | -- | 1.8 ± 0.1 | 307.0 ± 3.7 | 298.0 ± 3.6 | -46.0 ± 0.2 | 0.5182 ± 0.0273 | 0.3731 ± 0.0292 | 0.5742 ± 0.0455 | -- |
| PB2-05212022 | PCEGS-163 | 202201454 | -- | 2.3 ± 0.1 | 307.4 ± 3.7 | 298.4 ± 3.6 | -46.0 ± 0.2 | 0.5364 ± 0.0315 | 0.3922 ± 0.0335 | 0.6045 ± 0.0521 | -- |
| PB2-06022022 | PCEGS-169 | 202201459 | -- | 2.3 ± 0.1 | 307.3 ± 3.7 | 298.3 ± 3.6 | -40.3 ± 0.2 | 0.4920 ± 0.0291 | 0.3486 ± 0.0312 | 0.5371 ± 0.0486 | -- |
| | | | | | | | | | *Mean ± 1σ (All blanks)* | *0.5595 ± 0.0371* | |
| | | | | | | | | | *Mean ± 1σ (145,163 only)* | *0.5894 ± 0.0214* | |

Notes

1    Purdue Carbon Extraction and Graphitization System
2    Prime Lab ID
3    Mass graphitized for AMS analysis after small aliquot (ca. 9 ug C) taken for stable C isotopic analysis offline
4    Measured relative to OX-2 standard
5    Corrected for mass-dependent graphitization blank (based on AMS Split Mass C) and stable C composition
6    Sample values calculated using Diluted Mass C and corrected for mean procedural blank (All blanks)

**Replies to reviewer 2, Nicolás Young**

This manuscript presents a series of in situ 14C measurements in bedrock surfaces with the goal of assessing their reliability in 1) constraining the rate/timing of landscape emergence, and 2) constraining the timing of local deglaciation. The manuscript is set up by pointing out that the author's field area already has a pretty well constrained emergence curve and an independent constraint on the timing of local deglaciation (some mixture of traditional 14C and 10Be from Stroeven et al., 2016). Thus, the author's hope that their in situ 14C measurements "match" the independent controls as a test of their usefulness in this type of glacial setting. The approach here is to present a new tool to the community and show how it might work/behave in an ideal setting. By and large the in situ 14C measurements here match the independent record….things appear to have "worked."

I really enjoyed reading this manuscript. I think the authors set up the study nicely, gave the appropriate amount of background information, kept it shirt and focused, and didn't try to turn this dataset/manuscript into something it's not. Well done. Dataset and measurements look solid. This manuscript will certainly be of interest to the cosmogenic nuclide community, in particular users of in situ 14C. I think this manuscript is perfect for Ghron.

I do not have many suggestions here; again, this manuscript is already a nice little package that presents a clean story. I have a few minor comments, and then one more major comment. The latter is (likely) more directed at Nat and is the product of a timely independent conversation I have been having with Nat over the last week. The authors should pay particular attention to this and make sure it gets fixed not only for this manuscript but in relevant databases. Even though this is a more major comment, it should not affect the overall conclusions of this manuscript here.

**Thanks for your positive and helpful feedback on our manuscript.**

Major comment:

Authors did a great job at explaining how in situ 14C ages were calculated and also provide a great figure (Fig. 2) that shows the 14C calibration datasets that go into calculating the spallation 14C production rate of "13.35+-1.13" atoms/g/yr. Glancing at panel A in Figure 2 I thought that the New Zealand and Greenland calibration numbers looked a little high relative to the other calibration datasets. These calibration 14C concentrations, in addition to all the others, were recalculated using the methods of Hippe and Lifton, 2014. For the most part, this results in a very small adjustment to the 14C concentrations. However, for the New Zealand measurements the change is significant larger. The recalculated concentrations are ~6-16% higher. This was brought to my attention last week and we have had an email chain going on trying to figure out what was happening. It appears that this issue has been identified – the primary culprit was that in their original publications only the Fraction Modern was reported *instead* of the 13C-corrected Fraction Modern, so when the un-corrected Fraction Modern values were used in recalculation using the Hippe and Lifton, 2014 methods, its resulting in significantly higher sample 14C concentrations.

I looked through the code that the authors use here for calculating in situ 14C ages, and I couldn't find the individual calibration numbers, I did see the "13.35+-1.13" atoms/g/yr" in the constants file. But I searched around on the calculator website that the authors reference and

found this file, which I believe is what the authors are using to derive the global in situ 14C production rate: https://expage.github.io/data/prodrate/P-202306-input.txt I scrolled down to the 14C concentrations and, yes, what is here are the significantly higher New Zealand and Greenland concentrations. I also think these are the concentrations posted in the ICE-D database.

I raise this issue because those two datasets comprise ~15% of all the calibration measurements. With the pending concentration changes, the global production rate used here is going to get lower, and the uncertainty will likely improve. This will change the absolute in situ 14C ages in this manuscript a bit, but I believe with the error bars, all the in situ 14C ages will still overlap with the independent constraints (e.g., Fig. 3). The calibration concentrations need to be updated here, including in the relevant databases, and then the absolute ages need to be updated.

**Thanks for identifying this important issue, which we have now remedied. Co-author Lifton corresponded with Drs. Young and Schimmelpfennig (first authors of the manuscripts detailing the Greenland and New Zealand calibration datasets, respectively) and identified the main issues responsible for the discrepancy – 1) the published Fm values were not corrected for stable C isotopic composition, and 2) the more significant correction for the mass-dependent graphitization blank was not included in the published Fm values, nor was the formula used to make such corrections published. We have now calculated the corrected concentrations for those two calibration datasets, and have recalculated the global production rate to 12.81 at g$^{-1}$ yr$^{-1}$ – a ca. 4% decrease from 13.35 at g$^{-1}$ yr$^{-1}$. Updated concentrations for those two calibration sites are now also updated in the ICE-D Calibration database, and a corrigendum is in preparation for the Koester and Lifton (2023) citation from which the original recalculated data were taken.**

Minor comments:

Lines 80-81 – what about marine terraces? I think that's a common term/setting for developing emergence curves.

**There are no marine terraces here. Terrace formation is inhibited because the landscape is formed on tough, glacially scoured, crystalline bedrock (with only a patchy till cover), it's a highly fetch-limited environment, which limits wave energy necessary to terrace formation, and rapid isostatic rebound may have led to insufficient time for a terrace to form at a particular relative sea level.**

Lines 142-144 – wondering how applicable this 200m marine limit located 100 km away is to your site? Are there any closer marine limit benchmarks? Doesn't really change anything, just curious.

**We have located sites above the highest postglacial shoreline that are as close as possible to our Forsmark sites that were all located below sea level following deglaciation of the last ice sheet. Because the landscape is as low lying and low relief as it is, the nearest sites are about 100 km west. Once we found locations that were above the highest postglacial shoreline, we then had to locate outcrops in bedrock suitable for *in situ* $^{14}$C analyses, which has some additional minor influence on distances from Forsmark.**

Lines 225-235/Figure 2 – see main comment above

**Noted!**

Lines 398-403 – Agree with everything here. Not saying it's required, but if the authors wanted, it would be pretty simple to model a few endmember scenarios of MIS 3 exposure/MIS 2 cover +erosion histories to give the reader a sense on what it would take to arrive at the measured concentrations. For example, what would the evolution of 14C concentrations look like if you dosed the surface for 10 kyr during MIS, cover it during MIS 2 with no erosion, then re-expose at ~10 ka? Maybe just a few of these using *reasonable* exposure/burial/erosion rate constraints to give the reader how in situ14C concentrations may have evolved through the latter half of the glacial cycle?

**Thanks for this good suggestion. We have added a new figure, "Figure 5", that shows the hypothetical development of $^{14}$C concentration in the five samples above the highest shoreline in two different ice cover scenarios. We assume no glacial or interglacial erosion, continuous exposure during ice-free periods, and full shielding from cosmic rays during periods with ice cover. The simulations are done over the last 80 ka with ice cover during the periods 70-57 and 35-10.7 ka BP for the longer ice cover scenario and 66-60 and 28-10.7 ka BP for the short ice cover scenario. The simulations clearly show that if the MIS-2 ice advanced over the samples at 28 ka or earlier, the pre-MIS-2 ice history does not matter much for the present-day $^{14}$C concentration.**

Replying to my own review here, just want to be clear. The main issue is that the 13C **+ Blank** corrected Fraction Modern value needed to be used for the 14C calibration samples I mentioned. Sorry for the confusion.

**Thank you again for clarifying this, we have adjusted the data and figures accordingly – see response above.**

**Replies to the Associate Editor report**

Thank you for your editorial comments. We have given them careful consideration and prefer to keep Figures 2 and 4 as they are for reasons detailed below.

We prefer to keep the production rate calibration (illustrated in Figure 2) as it is because:

(i)     We exclude samples that are statistical outliers so that the site production rates match our assumption that all samples of the production rate calibration sites have experienced a single continuous period of full exposure to cosmic rays.

(ii)    The outlier rejection is done for each site because we want each site to have a reasonably well-clustered production rate. The global production rate is then calculated as an arithmetic mean of all site production rates.

(iii)   Cosmogenic nuclide production rates have generally been calculated using methods like those we have used, including outlier rejection and averaging of production rates from multiple samples/sites.

(iv)    Our method is similar to the calculator v.3 method (https://sites.google.com/a/bgc.org/v3docs/documentation-v3-exposure-age-calculator/4-ancillary-calculations-and-plots):
"First, compute the reference production rate implied by each measurement in the calibration data set. Average these at each site. Then, take the average of the site averages."
We have done this, but with the following differences:
  - We use a consequent and predictable outlier rejection method.
  - We calculate site production rates as weighted means because we have uncertainties on the sample production rates and can/want to take those uncertainties into account.
  - We use a slightly different approximation of the global uncertainty to take the sample uncertainties into account.

We prefer to keep the normalized kernel density estimates shown in Figure 4 because:

(i)     Normalized kernel density estimate plots are commonly used in the cosmogenic nuclide dating literature.

(ii)    Problems relevant to fission track ages, which may justify the use of radial plots, are largely irrelevant to our ages based on cosmogenic nuclides given that we have five specific ages with individual uncertainties assumed to be drawn from one normal distribution. We simply use the plot of normalized kernel density estimates to illustrate the spread of ages and their overlap with the reconstructed deglaciation ages.

(iii)   We strongly suspect that displaying our data as radial plots would not make it easier for scientists working with cosmogenic nuclide dating to understand what we are trying to show. Indeed, we would be more confused, rather than helped, by a radial plot of these data.

**Rely to further comments from the Associate Editor.**

We thank the associate editor for his further correspondence, which included guidance to change the compilation of Figure 4. This provoked discussion within our group regarding pdf's (camel plots), what they are if they are not pdfs, and whether they are an appropriate way to present dating results. We did not arrive at a consensus. We accept the associate editor's rebuttal of our point (i) above that just because the cosmogenic dating community commonly presents data as 'pdf's, this neither demonstrates their correctness nor justifies their use. We also greatly appreciate the associate editor's efforts to drive high scientific standards. The downside to the new Figure 4 is that it unfortunately does not look as good as the original 'camel plot', which was more effective in clearly conveying our data. However, we accept that scientific correctness outweighs artistic merit.